# Climate and urbanization drive changes in the habitat suitability of Schistosoma mansoni competent snails in Brazil

Caroline K. Glidden [1,2] ✉, Alyson L. Singleton[3], Andrew Chamberlin [4], Roseli Tuan[5], Raquel G. S. Palasio [5], Roberta Lima Caldeira[6], Antônio Miguel V. Monteiro[7], Kamazima M. M. Lwiza[8], Ping Liu[8], Vivian Silva[7], Tejas S. Athni [9], Susanne H. Sokolow[10,11], Erin A. Mordecai [1,2] & Giulio A. De Leo [4]

Schistosomiasis is a neglected tropical disease caused by *Schistosoma* parasites. *Schistosoma* are obligate parasites of freshwater *Biomphalaria* and *Bulinus* snails, thus controlling snail populations is critical to reducing transmission risk. As snails are sensitive to environmental conditions, we expect their distribution is significantly impacted by global change. Here, we used machine learning, remote sensing, and 30 years of snail occurrence records to map the historical and current distribution of forward-transmitting *Biomphalaria* hosts throughout Brazil. We identified key features influencing the distribution of suitable habitat and determined how *Biomphalaria* habitat has changed with climate and urbanization over the last three decades. Our models show that climate change has driven broad shifts in snail host range, whereas expansion of urban and peri-urban areas has driven localized increases in habitat suitability. Elucidating change in *Biomphalaria* distribution —while accounting for non-linearities that are difficult to detect from local case studies—can help inform schistosomiasis control strategies.

Schistosomiasis is a globally distributed, debilitating, and sometimes fatal, disease of poverty caused by *Schistosoma* blood flukes: parasitic flatworms (class:Trematoda) of freshwater snails and vertebrates[1,2]. Schistosomiasis currently affects over 250 million people in tropical and subtropical regions of the Americas, Africa, and Asia[1,2]. With more than 800 million people living in regions at risk for transmission, schistosomiasis is one of the most important neglected tropical diseases, second only to malaria in disease burden. In the Americas, Brazil shoulders the largest burden of this disease with 2–6 million people currently infected[3]. The World Health Organization includes

schistosomiasis in the new neglected tropical disease roadmap for elimination and control by 2030[4]. However, changes in the distribution of the disease have made areas in need of public health interventions a moving target[5], thus investment in adaptive control efforts is increasingly needed to make progress towards this goal.

A prerequisite for schistosomiasis transmission is the presence of the obligate intermediate host of the parasite: freshwater snails of the genus *Biomphalaria* (Preston, 1910)---hosts for *Schistosoma mansoni*, the parasite causing intestinal schistosomiasis—and *Bulinus* (O.F. Müller, 1781), hosts for *S. haematobium*, the parasite responsible of

[1]Stanford University, Department of Biology, Institute for Human-Centered AI, Stanford, CA, USA. [2]Stanford University, Department of Biology, Stanford, CA, USA. [3]Stanford University, Emmett Interdisciplinary Program in Environment and Resources, Stanford, CA, USA. [4]Stanford University, Department of Oceans, Hopkins Marine Station, Pacific Grove, CA, USA. [5]Pasteur Institute, São Paulo, Brazil. [6]Fiocruz Minas/Belo Horizonte, Minas Gerais, Brazil. [7]National Institute for Space Research, São José dos Campos, Brazil. [8]Stony Brook University, Stony Brook, New York, NY, USA. [9]Harvard Medical School, Boston, MA, USA. [10]Stanford University, Woods Institute for the Environment, Stanford, CA, USA. [11]Marine Science Institute, University of California, Santa Barbara, CA, USA. ✉e-mail: cglidden@stanford.edu

uro-genital schistosomiasis. In Brazil, an endemic country for intestinal schistosomiasis, *S. mansoni* is transmitted via three species of *Biomphalaria* snails: *Biomphalaria glabrata* (Say, 1818), *B. straminea* (Dunker, 1848), and *B. tenagophila* (D'Orbigny, 1835)[6]. Therefore, the distribution and abundance of the snail host, at least in part, underlies the observed spatiotemporal variation in schistosomiasis cases where focal transmission occurs. As ectotherms, temperature strongly influences fundamental snail biological processes, including growth rates and reproductive rates; as freshwater animals, precipitation can influence habitat availability and phenology, all of which ultimately impact snail fecundity and survival[7,8]. Schistosomiasis has historically been considered a rural disease. However, in the 1990s, at the start of our study, schistosomiasis began to emerge in some urban areas, such as the Recife Metropolitan area in Pernambuco, Brazil[9]. Since then, a growing portfolio of research in Brazil has demonstrated that *Biomphalaria* snails can thrive in human dominated environments within urban and peri-urban areas, such as drainage ditches, irrigation systems for small-scale agriculture, and unpaved flooded roads, all of which are often found in areas with marginalized populations and informal settlements with limited access to clean water, sanitation, and wastewater treatment[10–15]. With climate change and land-use change on the rise, we expect that the extent of snail habitat suitability has shifted through time and will continue to shift with ongoing global change—continuing to move the target for public health interventions. As such, understanding the spatial distribution of the intermediate host snails and their environmental, ecological, and socioeconomic determinants is a public health priority as it facilitates dynamic and precise identification of transmission hotspots to prioritize snail surveillance and removal.

Brazil aims for schistosomiasis elimination by 2030 and indeed has reduced transmission substantially since the inception of their national surveillance system in the 1950s[16]. However, total elimination remains elusive. Moreover, temporal trends in schistosomiasis related deaths have remained stable from 1999 to 2018 throughout most of the country, and has increased in the Northeast, where *B. glabrata* and *B. straminea* are found[17]. Globally, schistosomiasis control has been orchestrated through preventative and reactive medical treatment via administration of the drug praziquantel[2], WASH interventions (access to safe water, sanitation, and hygiene)[2], and environmental interventions (e.g., water engineering and snail control)[18,19]. In Brazil, public health professionals have found that mass drug administration has temporary effects[16]. Therefore, current recommendations for risk reduction include integration of water sanitation, community-education, and removal of snail breeding sites through non-molluscicide based environmental interventions, such as aquatic vegetation removal, draining flooded areas, and modification of watercourses. Mass drug administration is only suggested under exceptional circumstances[16]. Elucidating how environmental variables have influenced past changes in snail occurrence can help to anticipate which areas may benefit from increased surveillance for disease emergence and facilitate more precise allocation of resources.

Past efforts have mapped the current climate envelope of snail habitats at the national and regional scale as well as projected the climate envelope of *Biomphalaria* under future climate conditions[20,21]. The last publication of a national model of Brazil in 2012 included climate variables and vegetation greenness to map the static distribution of *Biomphalaria*[20]. However, other environmental characteristics critical to snail habitat, such as land-use, have yet to be incorporated into snail habitat suitability maps. Since then, remote sensing climate and environmental data has improved in both quality and scale (i.e., it is available at finer temporal and spatial resolutions). In particular, fine spatial and temporal resolution (e.g., 30 m 1 km; annual) land-use/land-cover, human population density, and hydrological variables are now available at the continental to global scale[22,23]. An increased number of spatially (100 m) and temporally precise

(annual) georeferenced biodiversity data for Brazil, including presence of *Schistosoma* parasite-competent snails, are now available from several research databases and public collections. Finally, advances in machine learning, particularly the accessibility of tree-based machine learning models, allow us to not only identify important variables, which is typical of machine learning models, but also to assess the functional form of the relationship between the environmental variable and habitat suitability (e.g, habitat suitability increases with precipitation seasonality)[24,25], which was not explored in previous studies mapping snail habitat suitability. Leveraging this new wealth of data and interpretable yet innovative machine learning algorithm, it is now possible to develop species distribution models (SDMs) of *Biomphalaria* snails across multiple time steps across a large geographical extent—all of Brazil—but also at a fine-scale spatial resolution (1 km$^2$), which was not possible ten years ago.

The goal of this paper is to map the historical and current distributions of the three competent snail intermediate hosts of *Schistosoma mansoni* in Brazil using a wide range of remotely sensed climatological and land-use data, a cutting edge, powerful and highly flexible machine learning model (Extreme Gradient Boosted Regression Trees), and a unique data set of over 11,000 georeferenced records of *Schistosoma* parasite competent snail occurrences spanning over three decades. Specifically, by identifying environmental variables that most contributed to the probability of snail occurrence and using counterfactual analyses, we asked: (i) what are the key features of the environment, and their functional response, that influence snail habitat suitability? And (ii) how has snail habitat suitability shifted with climate and urbanization throughout the last three decades? By doing so we develop a nation-wide consensus on the relationship between the environmental variables and snail habitat (e.g., determining if snail habitat is closer to rural or urban areas) and illuminate how dimensions of global change have and will shift targets for snail management.

In contrast to projections of snail distributions under future climate change, which are difficult to validate, the long time series of snail data allowed us to (i) use hindcasting—i.e., the process of testing statistical models by comparing them to actual historical observations to determine how well the models match the historical record—to validate SDMs' precision on the basis of actual field observations and (ii) counterfactual analysis to determine the relative importance of specific climatological and land-use changes in explaining the observed geographical range shift of *Schistosoma* parasite competent snails. This work will help to formulate more reliable hypotheses about how snail habitat suitability may change under current environmental trajectories allowing us to evaluate whether future projections are in line with what we expect based on the observed change through time.

In general, this study is one of the first to examine the compounding impacts of climate and land-use change on the distribution of infectious disease hosts at a national scale. Ultimately, it helps to predict the potential outcomes of interactions among local (e.g., urbanization) and large-scale (e.g., climate) environmental factors on environmentally transmitted disease risk.

## Results
### Model validation & hindcasting performance
We used spatial cross-validation to evaluate model performance. Model performance was evaluated using sensitivity (i.e., the proportion of occurrences the model correctly identifies as occurrences), model specificity (i.e., the proportion of background points that the model correctly identifies as background), and model area-under-the-curve (AUC; i.e., a measure that calculates how well the model correctly distinguishes occurrence points from background points, where AUC $\leq$ 0.5 indicates the model performs no better than a coin flip). Performance was high for each snail species: mean AUC and sensitivity was $\geq$0.80 and model sensitivity was $\geq$0.80 for all species. Mean model specificity was $\geq$0.70 for all species (Table 1; supplement p 7). Model

performance was also high for the hindcasting analysis, where we used the trained model to predict out-of-sample data collected between 1990 and 1999. For all species, AUC was >0.80 (*B. glabrata* = 0.89; *B. straminea* = 0.87; *B. tenagophila* = 0.81) and sensitivity was >0.80 (*B. glabrata* = 0.93; *B. straminea* = 0.82; *B. tenagophila* = 0.89) (Table 1; supplement p 7).

### Table 1 | Model performance metrics for the three *Biomphalaria* species

| Species | No. occurrence points | AUC | Sensitivity | Specificity |
|---|---|---|---|---|
| *Model validation: 5-fold spatial cross-validation* | | | | |
| *B. glabrata* | 165 | 0.85 (0.76–0.94) | 0.85 (0.79–0.92) | 0.80 (0.71–0.89) |
| *B. straminea* | 283 | 0.91 (0.88–0.95) | 0.85 (0.74–0.95) | 0.86 (0.79–0.92) |
| *B. tena-gophila* | 173 | 0.80 (0.64–0.95) | 0.92 (0.85–0.99) | 0.66 (0.40–0.92) |
| *Hindcasting: testing model on 1990–1999 data* | | | | |
| *B. glabrata* | 40 | 0.83 | 0.75 | 0.77 |
| *B. straminea* | 33 | 0.87 | 0.82 | 0.86 |
| *B. tena-gophila* | 28 | 0.83 | 0.93 | 0.68 |

The results for 5-fold cross validation include the mean, with the 95% confidence interval in parentheses. For model validation (5-fold cross validation), the number of background points were 2x the number of occurrence points. The number of background points used for hindcast testing, which were based on retaining one point per 1 km² grid cell, was 1688, 1691, and 1684 for *B. glabrata*, *B. straminea*, and *B. tenagophila*, respectively. Notably, these points were not used to train the model but only to test model predictions, therefore, class imbalance does not impact results.

## Feature contribution and functional response

Across the three species we found that, on average (i.e., across bootstrapping iterations), the features associated with climate, urbanization, and agricultural land-cover contributed the most to the predicted probability of snail occurrence, as opposed to features associated with hydrology or soil properties (Figs. 1–3).

For *B. glabrata* and *B. straminea*, we found that four out of the top five predictors of snail occurrence were climate variables, with three of these variables associated with precipitation (precipitation seasonality, precipitation in the driest month, and precipitation in the wettest quarter) and the fourth variable associated with temperature (isothermality) (Figs. 1a and 2a). Our model indicates that, for both species, snail habitat suitability peaked at high precipitation seasonality (12 CV, i.e., the coefficient of variance of monthly precipitation over one year) but decreased in areas where, on average, the mean monthly precipitation amount in the wettest quarter was high (<6000 kg m$^{-2}$ year$^{-1}$) (Figs. 1b and 2b). For *B. glabrata*, habitat suitability also decreased when precipitation amount was low in the driest month of the year (<500 kg m$^{-2}$ year$^{-1}$) (Fig. 1b), while for *B. straminea*, habitat suitability decreased when average annual precipitation amount was high (<6000 kg m$^{-2}$ year$^{-1}$) (Fig. 2b). In regard to temperature, *B.glabrata* and *B. straminea* habitat suitability non-linearly increased with isothermality, with respective peaks at 6 and 7 °C (ratio of diurnal variation in relation to annual variation in temperature). In accordance with the known biology of *B. tenagophila*, which has lower thermal limits for survival and reproduction, the climate profile for *B. tenagophila* significantly differed from that of *B. glabrata* and *B. straminea*. Climate variables made up only two of the top five predictors (Fig. 3a). *B. tenagophila* habitat suitability non-linearly decreased with mean daily air temperature during the

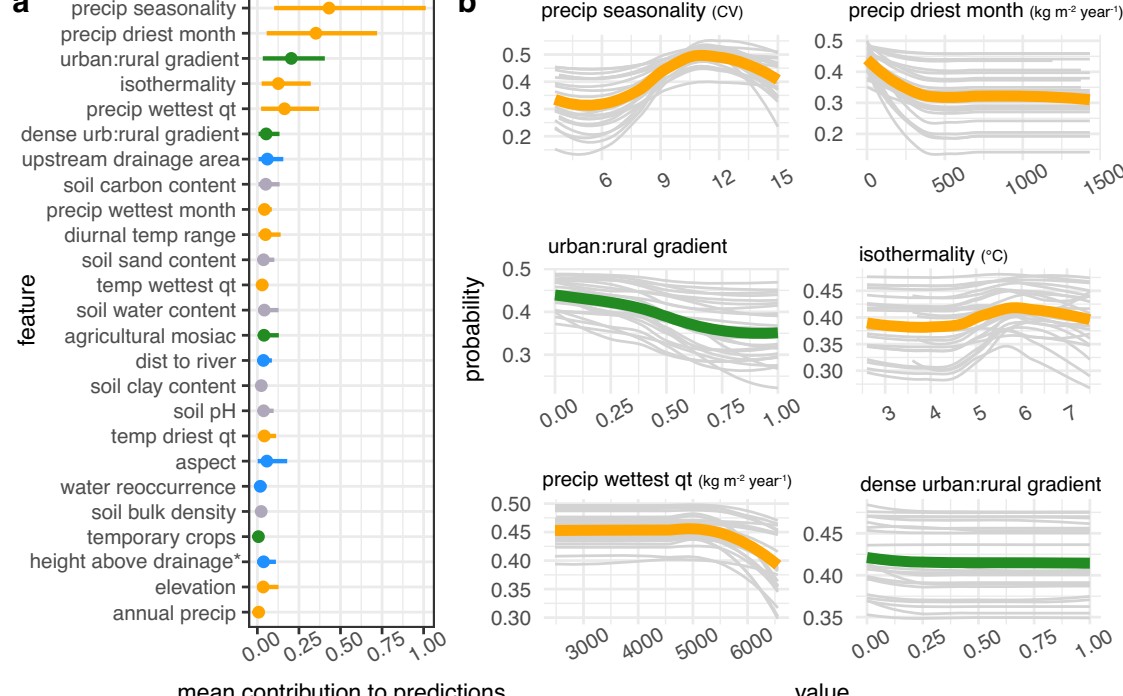

**Fig. 1 | *B. glabrata* habitat suitability is most sensitive to climate and urbanization variables. a** Feature contribution for each feature. Points are the absolute value of the mean contribution to predictions (mean |SHAP Value|) for the covariate across all data points (i.e., global feature contribution), bars represent the 95% confidence interval across 25 bootstrapping iterations. **b** Response of snail habitat suitability (partial dependence plots) to top six contributing variables. For each univariate plot, light gray lines represent the probability of occurrence after controlling for all other covariates for each bootstrapping iteration and green (land-use variable) or orange (climate variable) represent the average across 25 bootstrapping iterations. *height above nearest drainage (vertical distance to nearest stream surface or bed).

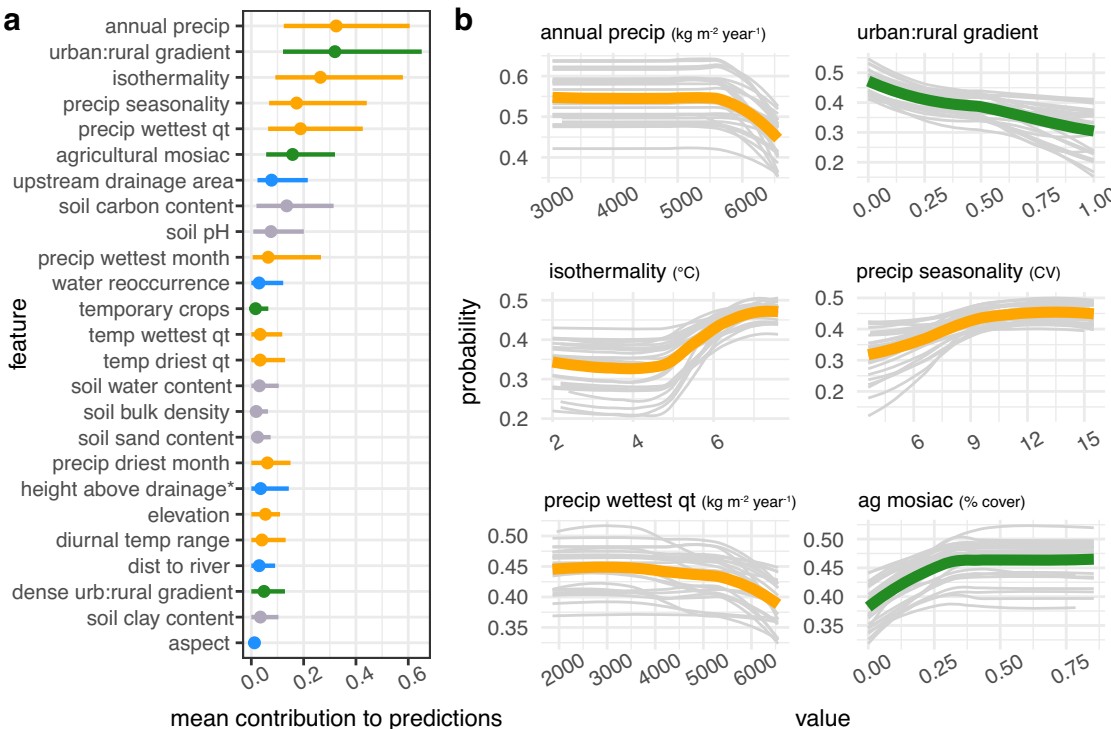

**Fig. 2 | *B. straminea* habitat suitability is most sensitive to climate and land-use. a** Feature contribution for each feature. Points are the absolute value of the mean contribution to predictions (mean |SHAP Value|) for the covariate across all data points (i.e., global feature contribution), bars represent the 95% confidence interval across 25 bootstrapping iterations. **b** Response of snail habitat suitability (partial dependence plots) to top six contributing variables. For each univariate plot, light gray lines represent the probability of occurrence after controlling for all other covariates for each bootstrapping iteration and green (land-use variable) or orange (climate variable) represent the average across 25 bootstrapping iterations. *height above nearest drainage (vertical distance to nearest stream surface or bed).

driest quarter of the year (>20 °C) and peaked when precipitation seasonality is low (7 CV) (Fig. 3b).

Globally, schistosomiasis is considered a rural disease but has been observed close to or in urban areas in Brazil. To measure the impact of urbanization on *Biomphalaria* occurrence, and because an urban versus rural dichotomy does not capture the complexity of urbanization in Brazil[26], we derived an urban to rural gradient, where 0 indicated that the snail was observed in an urban area and 1 indicated the snail was in a rural area, and continuous values from 0-1 indicated the relative location of the snail along this gradient. For all three snail species, we found that location along the urban (>300 per people per km, with 2500 people in a contiguous area) to rural (<300 people per 1 km²) gradient was among the top five predictors, with location along the high density urban (>1500 per people per km² and >150,000 in a contiguous area) to rural gradient also among the top five predictors for *B. tenagophila* (Figs. 1a, 2a and 3a). Snail habitat suitability increased for all snails within pixels closer to urban areas (0 = in an urban area) (Figs. 1b, 2b and 3b). The relationship between *B. tenagophila* and the urban to rural gradient followed a pattern of exponential decay, whereas the change in probability was more gradual for *B. glabrata*, and nearly linear for *B. straminea*. Agricultural crop cover and temporary crop cover were among the top six predictors for *B. straminea* and *B. tenagopila*, respectively, with habitat suitability increasing with % area of crop cover (Figs. 2 and 3).

**Changing snail distribution**
According to model predictions, we found that habitat suitability of *B. glabrata* decreased from 1992 to 2017 throughout western Brazil and increased towards coastal areas of the Southeast and Northeast region (Fig. 4a–c). In contrast, we found that habitat suitability of *B. straminea* primarily expanded southward, whereas for *B. tenagophila* suitability contracted within the state of São Paulo (Fig. 4d–i). In total, our model

predicted a fairly large change in habitat suitability for each *Biomphalaria* host species: 58% (CI: 52–66%; CI is the 95% confidence interval estimated from a bootstrapping procedure) of the area of Brazil experienced a change in *B. glabrata* suitability, 55% (CI: 45–64%) for *B. straminea*, and 28% (CI: 19–34%) for *B. tenagophila* (Table 2; supplement p 23). For *B. glabrata*, the area that decreased in suitability was equivalent to the area that increased, whereas for *B. straminea* the increase in area of habitat suitability was greater than the decrease in area, and for *B. tenagophila* the decrease in the area of habitat suitability was marginally higher than the increase (Table 2; supplement p 23).

We conducted a counterfactual analysis to quantify the change in habitat suitability associated with change in climate and urbanization. In brief, our counterfactual analysis compared the observed change in habitat suitability to the change in habitat suitability that would have occurred if the climate or urban extent had not changed. This methodology allowed us to isolate the change in habitat suitability associated with the change in these different dimensions of global change. Based on our counterfactual analysis, for each species, large-scale changes in predicted habitat suitability were driven by regional variation in climate between the historical (1990–1999) and recent (2000–2020) time periods (Table 2; Fig. 5). For all three species, the climate-associated increases in suitability in some areas were offset by climate-associated decreases in suitability in others (Table 2; supplement p 23), indicating broad scale shifts as opposed to net increases and decreases in suitability. Throughout the last three decades, the greatest change in urbanization was the growth of small-medium sized urban areas (300–1500 people per km² with >2500 in a contiguous area) (supplement p 26). As such, in contrast to larger-scale climate-driven shifts, changes in the probability of snail occurrence associated with urbanization were highly localized at the outskirts of existing cities or new cities that emerged between the two time points (Fig. 6;

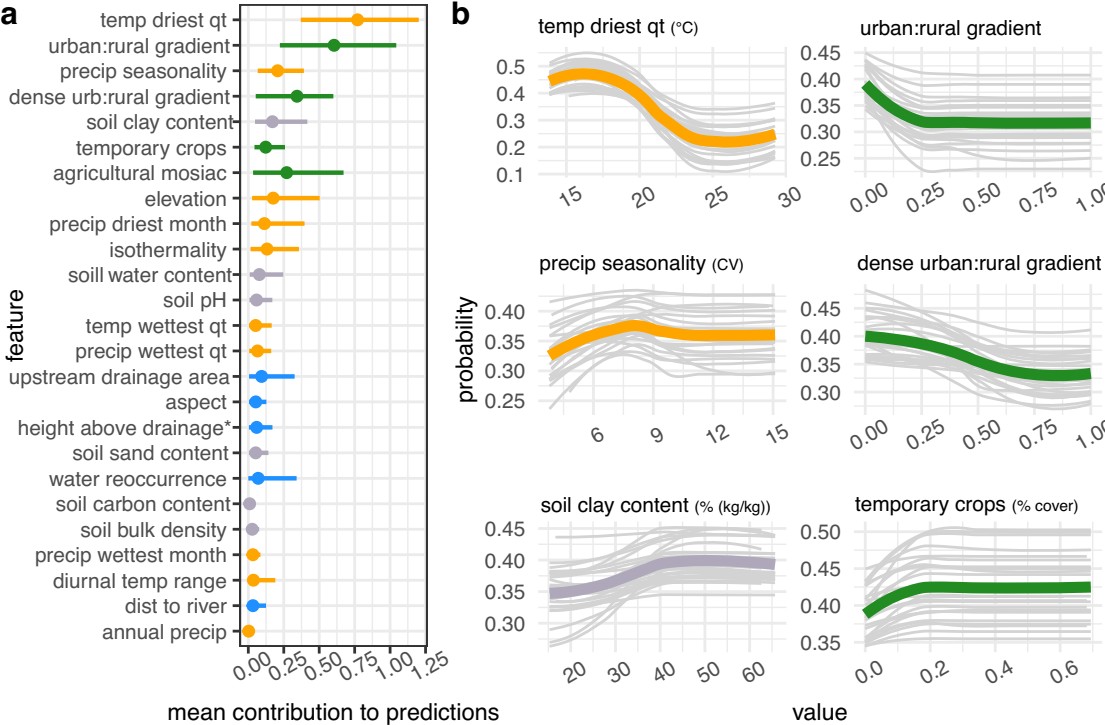

**Fig. 3 | *B. tenagophila* habitat suitability is most sensitive to climate, land-use, and soil clay. a** Feature contribution for each feature. Points are the absolute value of the mean contribution to predictions (mean |SHAP Value|) for the covariate across all data points (i.e., global feature contribution), bars represent the 95% confidence interval across 25 bootstrapping iterations. **b** Response of snail habitat suitability (partial dependence plots) to top six contributing variables. For each univariate plot, light gray lines represent the probability of occurrence after controlling for all other covariates for each bootstrapping iteration and green (land-use variable) or orange (climate variable) represent the average across 25 bootstrapping iterations. *height above nearest drainage (vertical distance to nearest stream surface or bed).

supplement pp 24), with a largely positive increase in suitability (Table 2; supplement pp 23–24). For example, in the Vale do Rio Doce, a meso-region, i.e., an administrative level between municipality and state, bordering Minas Gerais and Espírito Santo States where schistosomiasis is endemic, there were numerous small-medium cities that emerged (e.g., Baxio Gaundú, Inhapim), while the area of the dense cities (Ipatinga and Governador Valadares) hardly changed (Fig. 6a, d). As such, our model predicts that the most notable change in *B. glabrata* habitat suitability associated with urbanization is in the northeast and south of this meso-region. In the meso-regions of the Recife Metropolitan Area, an area where *B. straminea* has been found infected with *S. mansoni*[27], and São Paulo Metropolitan area, an area where *B. tenagophila* has been associated with foci of transmission[28], the most notable changes in *B. straminea* and *B. tenagophila* habitat suitability were at city outskirts where new urban areas had developed around the periphery of the cities (Fig. 6b, c, e, f).

## Discussion

Schistosomiasis snail intermediate host habitat suitability depends strongly on climate and land-use variation and, as a result, have shifted substantially in the last 30 years in Brazil (Figs. 4–6). Using remotely sensed data, machine learning, and a long-term dataset of expert-collected snail occurrence data, we mapped the distribution of snail habitat suitability at a fine spatial resolution (1 km²) and large geographic extent (national) with high accuracy (Fig. 4, supplement p 7). We found that climate and urbanization features are the most important predictors of *Biomphalaria* habitat suitability across all three species, after controlling for sampling bias (Figs. 1–3). Large-scale shifts in snail habitat suitability occurred due to regional changes in climatic variables, whereas urbanization influenced fine-scale, highly localized increases in habitat suitability within and around small-

medium sized urban areas perhaps due to lack of infrastructure in rapidly growing informal settlement, as people move from rural to urban areas both within and across Brazilian states[29] (Figs. 5 and 6, supplement pp 23–26).

With respect to climatological variables, snail habitat suitability was primarily influenced by precipitation patterns for *B. straminea* and *B. glabrata* and by temperature for *B. tenagophila* (Figs. 1–3). Counterintuitively, as snails inhabit freshwater bodies, *B. glabrata* habitat suitability was highest in areas with, on average, a low amount of precipitation during the driest month of the year (Fig. 1b). Low precipitation may create small pockets of surface water ideal for snail habitat, or, alternatively, concentrate *B. glabrata* water sources thus making them easier to find. Additionally, *B. glabrata* habitat suitability was highest in areas with high precipitation seasonality, i.e., a strong difference in precipitation between the dry season and the wet season (Fig. 2b). Perhaps freshwater habitats accumulate enough water during the wet season to support snail survival and reproduction throughout the dry season. Precipitation regimes have been rapidly changing, with Brazil currently experiencing a long and severe drought throughout regions of snail habitat[30]. As such, understanding the relationship between dry precipitation patterns and *B. glabarta* might be critical for controlling snail populations. *B. straminea* habitat suitability was high across a wide range of annual precipitation values but decreased at extremely high volumes (Fig. 2b), indicating it can tolerate most annual precipitation conditions but likely gets flushed out with intense precipitation. Further, *B. glabrata* and *B. straminea* habitat is most suitable at high values of isothermality -- with *B. straminea* habitat associated with higher isothermality than *B. glabrata* (Figs. 1b and 2b). Isothermality is highest in northern Brazil, as such this variable might best reflect the unique climate conditions found towards equatorial Brazil. Our model indicates that *B. tenagophila* has a distinct climatic niche as

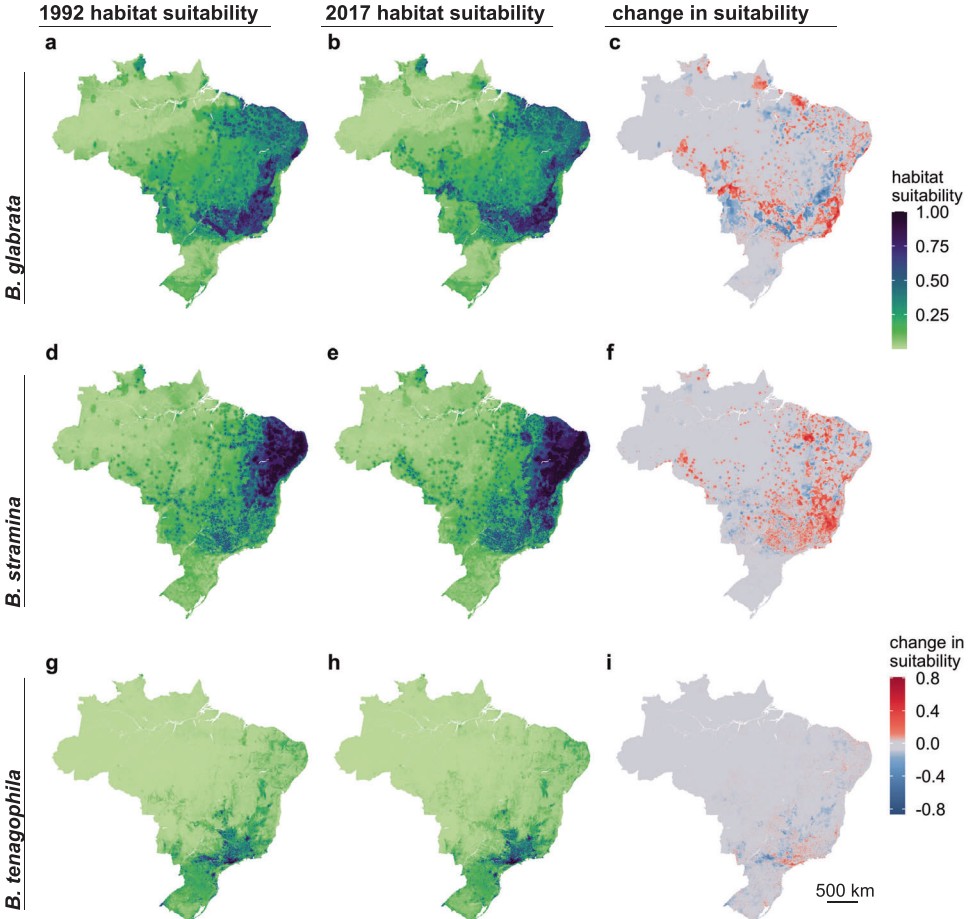

**Fig. 4 | Predicted habitat suitability of the three competent snail hosts shifted over 25 years.** The figure includes the distribution of each snail species in 1992 (**a** *B. glabrata*; **d** *B. straminea*; **g** *B. tenagophila*). and 2017 (**b** *B. glabrata*; **e** *B. straminea*; **h** *B. tenagophila*), where light green indicates low suitability and dark purple indicates high suitability. Panels (**c** *B. glabrata*; **f** *B. straminea*; **i** *B. tenagophila*) show the change in habitat suitability between the two time points, where red indicates an increase in habitat suitability and blue indicates a decrease in habitat suitability. Spatial resolution is 1 km². Figures are mean values per pixel across 25 bootstrapping iterations. For each species, predicted values are scaled from 0 to 1 using a min-max transformation.

it is found in areas with, on average, low mean daily temperatures during the driest quarter of the year, consistent with *B. tenagophila* almost exclusively found in the south of Brazil (Fig. 3b). Our results for *B. tenagophila* mirrored the previous national model of the *Biomphalaria* climate envelope in Brazil: mean monthly temperature of the driest quarter was one of the top predictors of snail occurrence[20]. However, our results contrast previously published results for *B. glabrata* and *B. straminea* as the previous national model found variables associated with temperature to be the strongest contributors to model predictions as opposed to precipitation[20]. This may be related to the mismatch of *Biomphalaria* occurrence points and the climatologies used in previous studies, which were retrieved from WorldClim that spans 1960–2000[31], whereas our climatologies overlap the decade the snail was collected. Yet, because this analysis is correlational, we cannot rule out the influence of unobserved variables that are correlated with these climatic conditions and may drive the discrepancy between the models. Further, it is difficult to comprehensively compare our results as the previous publication did not quantify the functional response between climate variables and probability of occurrence[20].

The location along the urban to rural gradient was among the top three predictors of habitat suitability for all three species (Figs. 1–3). The distance to a small-medium urban area (300–1500 people per 1 km² with >2500 people in contiguous pixels) was important for all three snail species, and distance to high density urban area (>1500

people per 1 km² with >150,000 people in contiguous pixels) was among the top five predictors of *B. tenagophila* habitat suitability. Although the rate of change in snail habitat suitability and urban to rural gradient differed between the three snail species, in general, habitat suitability for the three species is higher closer to urban areas than rural areas (Figs. 1–3). Case studies indicate that schistosomiasis, a disease that has been historically associated with rural areas, is urbanizing in Brazil[9,10], with empirical evidence supporting the hypothesis that urban and peri-urban areas provide the necessary abiotic factors for snail survival, while reducing the number of snail competitors and predators, ultimately increasing snail habitat suitability[10,32]. Small scale agricultural land-use found in peri-urban and urban areas[33]–agricultural mosaics and temporary crops—were top predictors of snail habitat suitability, providing some insight on the types of habitats that peri-urban and urban areas provide. However, as the location along the urban to rural gradient contributed more to model predictions than agricultural land-use and land-cover, there are likely other aspects of urban and peri-urban infrastructure that provide snail habitat, such as sewage systems that outflow into open ditches and unpaved streets in rapidly growing settlements[34], which also favor the dispersion in the environment of *S. mansoni* eggs with fecal waste[27,34]. For example[15], found that snails use sediment from used water and open-air sewage as a food source as opposed to aquatic plants. Critically, the population derived location along the urban to rural gradient is a rough proxy for estimating the impact of urban dynamics and should be updated in

future studies to better incorporate specific infrastructure (e.g., waste management and resulting prevalence of organic pollution) and land cover composition of urban and peri-urban areas and estimate edge effects, especially as they may lag behind change in population density and size.

Our models suggest that climate and urbanization have significantly contributed to a shift in *Biomphalaria* habitat suitability over the last three decades (Figs. 4 and 5, supplement 23–25). Distributions of *B. glabrata*, the most competent intermediate host, have shifted to coastal areas, particularly towards the southeast (Fig. 4), with our counterfactual analysis supporting the hypothesis that this shift is driven by both climate, particularly increased precipitation seasonality and drier dry seasons, and urbanization (Figs. 5 and 6, supplement p 24). Of the three host snails, the distribution of the habitat suitability for *B. straminea* has changed the most, with a large southward increase in habitat suitability, reflective of decreasing annual precipitation and increasing isothermality within Central West and southern Brazil across the two time points, and local changes associated with

### Table 2 | The proportion of the area of Brazil that is predicted to have experienced a change in habitat suitability between 1992 and 2017

| Predicted distribution (proportion of change in area) | | | |
|---|---|---|---|
| | Total change | Positive change | Negative change |
| *B. glabrata* | 0.58 (0.52–0.66) | 0.29 (0.36–0.34) | 0.29 (0.24–0.36) |
| *B. straminea* | 0.55 (0.4–0.64) | 0.31 (0.27–0.34) | 0.24 (0.16–0.30) |
| *B. tenagophila* | 0.28 (0.19–0.34) | 0.12 (0.08–0.16) | 0.25 (0.11–0.19) |
| Climate counterfactual (proportion of change in area) | | | |
| | Total change | Positive change | Negative change |
| *B. glabrata* | 0.54 (0.46–0.62) | 0.25 (0.21–0.30) | 0.29 (0.23–0.37) |
| *B. straminea* | 0.45 (0.33–0.56) | 0.24 (0.17–0.30) | 0.21 (0.12–0.29) |
| *B. tenagophila* | 0.18 (0.13–0.24) | 0.07 (0.04–0.10) | 0.11 (0.08–0.16) |
| Urban counterfactual (proportion of change in area) | | | |
| | Total change | Positive change | Negative change |
| *B. glabrata* | 0.16 (0.12–0.20) | 0.11 (0.08–0.14) | 0.05 (0.03–0.07) |
| *B. straminea* | 0.20 (0.15–0.24) | 0.14 (0.11–0.18) | 0.06 (0.04–0.07) |
| *B. tenagophila* | 0.04 (0.02–0.07) | 0.03 (0.05–0.04) | 0.01 (0.006–0.024) |

The counterfactuals indicate the percent of area change associated with those features. The table presents the mean values, with the 95% confidence intervals in parentheses across 25 bootstrap iterations.

urbanization (Figs. 2, 5, and 6, supplement pp 15–16). While *B. glabrata* is the most competent host, the expansive and moving habitat suitability of *B. straminea* suggests that this species may play a larger role in schistosomiasis persistence under a changing environment. In contrast, the geographic extent of *B. tenagophila* habitat suitability has decreased over the last three decades, most likely due to temporal variation in temperatures and reduced precipitation seasonality in Southeast Brazil (Figs. 2 and 5, supplement pp 15-16). These findings are in line with local *Biomphalaria* surveillance, which have observed that *B. straminea* has replaced *B. tenagophila* in some areas of São Paulo state[35].

While this analysis indicates large geographic shifts in snail habitat suitability, on-the-ground observations may not detect this shift because land-use, particularly urbanization, may create localized pockets that amplify or dilute snail habitat suitability in adjacent areas (Fig. 6; supplement pp 24-25). For example, around some new cities we observed increases, roughly 10 km wide, in suitable habitat associated with both climate and urbanization. For instance, the cities of Inhapim and Baixo Guandú grew from rural areas to small cities with >20,000 people between the historical and recent time periods. Both cities have reported some of the highest numbers of schistosomiasis cases between 2007 and 2017 in their respective states (Minas Gerais and Espírito Santo)[36,37]. Notably, our analysis shows that there has been a decrease in suitability associated with the "disappearance" of small-medium urban areas between the time points. This warrants further on-the-ground investigation and validation as this result may be reflective of change in infrastructure associated with population size or might be an artifact of our definition of urbanization, where the population size and/or density only marginally drops below our urban population size and density thresholds, thus showing up on maps as a dissolved urban area when in reality there has been little change in infrastructure that impacts schistosomiasis risk.

Overall, this analysis provides a fine scale yet bird's-eye view of changing snail habitat suitability, allowing us to tease apart these interactions and better understand change in snail distribution through time. These findings may guide future management and policy work, such as allocating resources to control schistosomiasis in urban and peri-urban areas, and monitoring snail occurrence at the edge of the described climate envelopes. An important next step is identifying the mechanistic underpinnings of snail occurrence in peri-urban and urban settings, which will enable even higher precision mapping of snail habitat and the development of ecological interventions that facilitate sustainable snail removal. For example, in Senegal[38], found specific species of aquatic vegetation that are strong

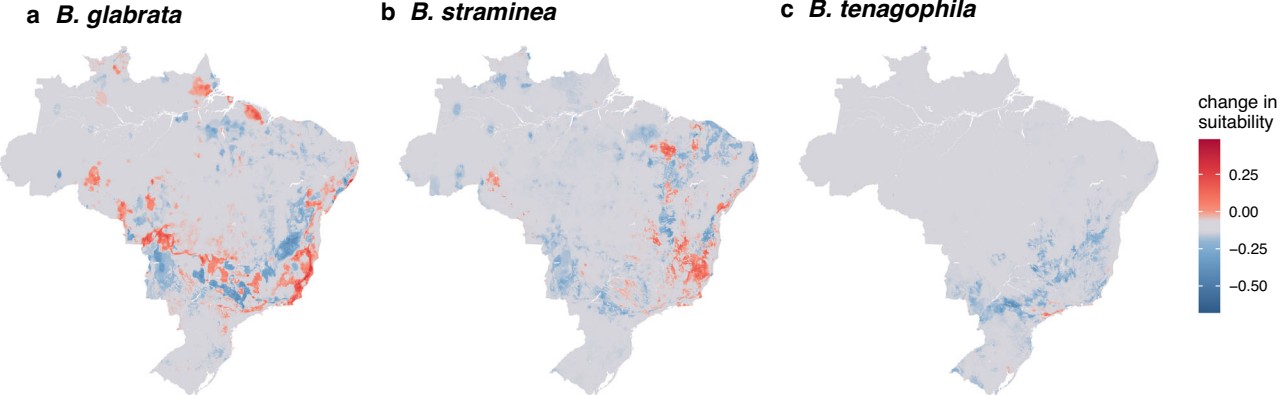

**a** *B. glabrata*  **b** *B. straminea*  **c** *B. tenagophila*

change in suitability
0.25
0.00
−0.25
−0.50

**Fig. 5 | Climate drives large scale changes in suitability for all three species.** The difference in distribution between the predicted distribution of 2017 and the counterfactual: the 2017 prediction estimated when holding climate features as they were observed in 1992. Panel **a** is for *B. glabrata*, **b** is for *B. straminea*, and **c** is for *B. tenagophila*. Red indicates the habitat that became more suitable with change in climate and blue indicates the climate became less suitable with the observed change in climate (i.e., the change in predicted distribution is due to the change in climate). Figures are mean values per pixel across 25 bootstrapping iterations.

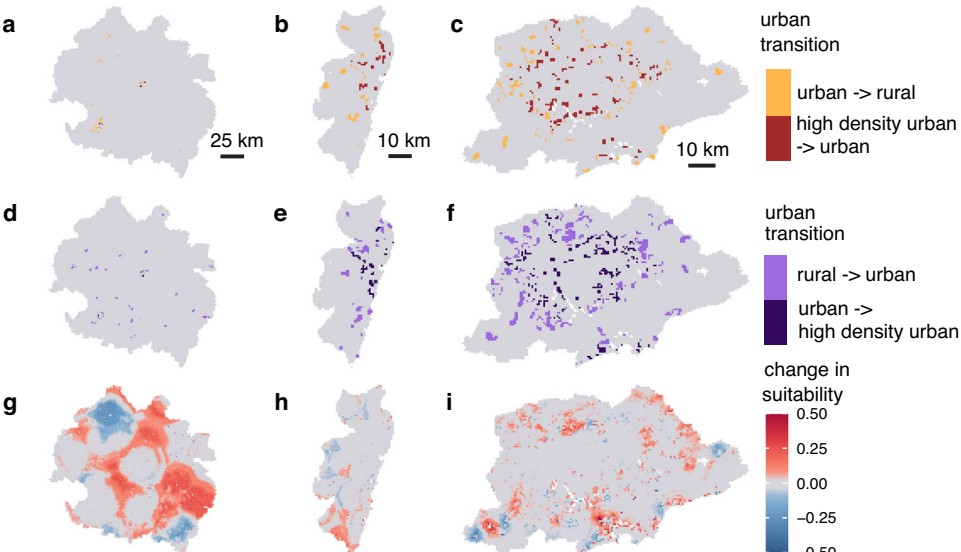

**Fig. 6 | An example of how urbanization affects habitat suitability on small scales.** The change in urban extent for the Vale do Rio Doce (**a**, **d**), Recife Metropolitan (**b**, **e**), and São Paulo Metropolitan (**c**, **f**) mesoregions between 1992 and 2017. **g**, **i** The difference in snail probability for 2017 when using urban-rural gradient values for 2017 versus 1992 (i.e, the change in snail probability attributed to change in urban distribution and extent) for *B. glabrata* in Vale do Rio Doce (**g**), *B. straminea* in Recife (**h**), and *B. tenagophila* in São Paulo (**i**). The Vale de Rio Doce region is much larger than the other two mesoregions, but the change in urban extent highlights that there was a significant amount of emergence of small-medium cities (300–1500 people per 1 km² pixel, with >2500 people in contiguous pixels) and this is associated with a significant change in habitat suitability. We chose the mesoregion for each snail based on evidence that the species is associated with foci of transmission and/or have been found infected with *Schistsoma mansoni* in that area. Figures **g**–**i** are mean values per pixel across 25 bootstrapping iterations.

predictors of *Bulinus* (the *Schistosoma* snail host in Senegal) abundance and schistosomiasis infection. Satellite imagery and deep learning models have been developed to identify this vegetation at <2-m resolution[39], providing a tool for highly targeted habitat control at a large geographic extent. Identifying fine-resolution biotic and abiotic drivers of snail habitat in peri-urban and urban Brazil will support similar precision mapping and resource-effective habitat removal. Importantly, *Schistosoma* parasites may have different environmental needs. Another necessary next step is to identify how snail habitat translates to human infection rates, and the environmental drivers of parasite presence that overlap with snail presence. This will enable identification of transmission hotspots as opposed to only snail habitat.

Beyond Brazil, our study may help to inform spatial-temporal variation of snail habitat at a global scale. For instance, *B. straminea* in South China, where it is an invasive species, has demonstrated similar changes in distribution over time: the snail's distribution has expanded away from the equator towards historically more temperate habitats[40]. Our results suggest that this pattern is likely due to changing climate, with localized suitability amplified or diluted by changing patterns in the extent of small to medium sized urban areas. Further, urban foci of transmission have emerged in Asia and Africa, such as in Ethiopia and Nigeria[10]. Our study emphasizes the need to evaluate the role urbanization might be playing in schistosomiasis persistence on a global scale.

Despite the three intermediate host snails having distinct socio-ecological niches within Brazil, our work underscores the importance of species-specific combinations of temperature, precipitation, and urban conditions that are highly predictive of snail occurrence, and therefore important indicators of schistosomiasis risk. Proactively anticipating shifts in snail habitat suitability and schistosomiasis risk is a major priority for targeting snail control and public health services and for mitigating disease emergence. Long-term snail surveillance and control programs face shifting demands as the seasonality, intensity, and geography of snail populations change with climate,

land-use, and socio-ecological conditions. Our study shows that snail habitat suitability has changed throughout the last three decades and provides profiles of baseline exposure risk that can help shift the spatial priorities of control efforts with changing socio-ecological conditions, land-use, and climate.

## Methods
We used machine-learning based (XGBoost) Species Distribution Models (SDMs), 30 years of expert-collected occurrence data, and remote sensing data to map the habitat suitability for each of the three *Schistosoma* parasite competent *Biomphalaria* species. SDMs use species occurrence coordinates and corresponding environmental variables to interpolate habitat suitability across large geographic extents. Machine learning SDMs are particularly flexible models that can handle non-linear relationships, collinear features (covariates), and complex interactions among features[25,41]. We trained the models on data collected from 2000 to 2020. Using data collected from 1990 to 1999 as a hold-out test set, we used a hindcasting approach to test the model.

### Occurrence points and data thinning
We used a combination of snail occurrence points collected by the Brazilian government programs, Medical Malacology Collection (CMM-Fiocruz) Minas Gerais and São Paulo Department of Public Health. In total, the programs collected 11,299 snail records that spanned 1990–2020 (supplement pp 1;8). For our analysis, for each focal species, we thinned data so that only one occurrence point was retained per 1 km² grid cell using the R package *dismo*[42]. Thinning data is essential to control for pseudo-replication[43].

### Background points
There is often a lack of true species absence data for model input and thus background points or pseudo-absences are used as a proxy for where the focal species does not occur[25]. To generate background points, we created a background mask using species records from the

Global Biodiversity Information Facility (GBIF)[44] as well as expert collected snails that were not the focal species. Specifically, from GBIF, we used georeferenced records from all freshwater animals in Brazil. We downloaded 55,057 coordinates, spanning 1990–2020, for 2135 species. Records from all freshwater animals allowed us to create a landscape of freshwater animal sampling locations and, thus, where snail species could plausibly have been sampled. These methods help to account for sampling bias, such as bias that may arise if sampling of ecological communities is concentrated around human population centers. We first overlaid a 1 km² resolution grid over Brazil. Each cell was then assigned a probability value based on the number of background points (records) within each grid cell. Using these probabilities, we selected the background points by probabilistically sampling twice the number of occurrence points from unique 1 km² grid cells. This procedure was repeated for each of the three competent host snail species.

### Environmental data processing

We included features related to climate, land-use/land cover (LULC), hydrology, topography, and soil properties in our distribution models (supplement pp 1–6). We used the CHELSA v2.1 monthly products to calculate 10-year climatologies spanning four decades (1980–1989, 1990–1999, 2000–2009, 2010–2017)[45]. The CHELSA dataset occurred at the coarsest spatial resolution: 1 km². To minimize issues with spatial autocorrelation, we aggregated all other spatial covariates to a spatial resolution of 1 km².

To estimate land-use and land-cover (LULC) variables, we used the MAPBIOMAS land-cover raster collection 7.0 v2[23] to calculate the percent cover of temporary crops (excluding monocultured crops: soy, sugarcane, rice, and cotton) and mosaic of use—two land-use types associated with free standing fresh water in urban and peri-urban farms —within the 1 km² grid cell during the year the data point was collected. We also incorporated hydrology and water availability using the Global Surface Water dataset[46], MERIT Hydro data (100 m resolution)[47], and WWF HydroSHEDS Free Flowing River dataset[48], topography using the NASA NASADEM Digital Elevation dataset (30 m resolution)[49], and the OpenLandMap dataset (250 m resolution) to calculate soil properties[50–54]. We removed highly correlated features ($r > 0.75$) (supplement pp 1–6).

### Urban-to-Rural gradient index

We defined urban and rural areas using population density and population size as a proxy. First, we defined an urban area based on an aggregation of urban definitions throughout South America: an urban area had population densities from 300 to 1500 people within 1 km² with contiguous pixels totaling at least 2500 people[55]. Next, we included high density urban areas in both analyses: >1500 people per grid cell with contiguous grid cells totaling more than 150,000 people[56]. As population-based definitions of urban areas are inconsistent, as a separate analysis, we repeated the analysis with alternative definitions of urbanization as described in supplement p 2, and the results yielded similar results (supplement pp 7, 17–22). We used WorldPop to estimate population size and cumulative cost mapping— where every pixel is assigned the total cost of the lowest cost path (distance) to an urban pixel—to estimate the distance to the nearest urban and rural pixels for points collected between 2000 and 2020. WorldPop data is not available before 2000; as such, we derived population estimates from 1990 to 1999 by linearly interpolating between the 2000 WorldPop raster and the Global Human Development Layer, Population Grid 1975 raster[57]. To calculate the final variable—location along the urban to rural gradient—we divided distance to an urban area by distance to a rural area and standardized it from 0 to 1 using a max-min transformation. All environmental data was processed and aggregated to 1 km² grid cells around each data point using Google Earth Engine[58] and the GEE Python API in Google Co-

Laboratory, with the exception of CHELSA, which was downloaded from their AWS repository[59].

### Extreme gradient boosted regression tree SDMs

We used extreme gradient boosted regression tree models, using package *Xgboost*[60], for each SDM as boosted regression models were determined to yield the best performance and interpretability for modeling snail habitat in Brazil[41]. Specifically, when compared to MaxEnt and Random Forest models, gradient boosted machine predictions of snail habitat suitability performed with comparable accuracy but better aligned with expert knowledge of *Biomphalaria's* known distribution[41]. Extreme gradient boosted regression is a machine learning algorithm that creates an ensemble of weak decision trees to form a stronger prediction model by iteratively learning from weak classifiers and combining them into a strong classifier (i.e., boosting). Xgboost is flexible in that it allows for non-linearity, both among features (i.e., interactions) and between features and the response variable, higher collinearity among features than traditional statistical models, and non-random patterns of missing data, characteristics frequently observed in ecological data. Spatial cross-validation (5-fold), using the R package *blockCV*[61], was used to estimate the model's out-of-sample predictive power. Out-of-sample testing ultimately tests our model's ability to generalize biological patterns as opposed to the ability to predict complex patterns of the training dataset. We used spatial cross-validation so to minimize inflating model performance values due to spatial autocorrelation of environmental predictors[61]. In brief, we first split our dataset into five distinct geographic regions (5 folds). We then trained our model on four folds and validated the model (tested model performance) on the hold out fold. We repeated this process until all five geographic regions were tested as the hold-out dataset. For each spatial-cv step, model parameters were tuned and trained using Bayesian optimization via the R package *rBayesianOptimization*[62]. Specifically, we tuned: the learning parameter ("eta"; controls how much information from a new tree will be used for boosting), maximum depth of the trees ("max_depth"), the minimum weight necessary to create a new node ("min_child_weight"), the fraction of data used to grow each tree ("subsample"), and the fraction of features used to train each tree ("colsample_bytree"). We tuned the model by using 5-fold cross-validation to select the hyperparameters that minimized the loss function. We controlled the balance of negative to positive weights by setting "scale_pos_weight" to 2 as there were 2x as many background samples (0) as presence samples (1). All other model parameters were set at default settings. Model sensitivity, model specificity, and model area-under-the-curve (AUC) were calculated using the R package *pROC*[63] and *caret*[64]. Model training and validation was conducted using data from 2000 to 2020.

To evaluate the average contribution of each covariate to model predictions, we first identified important features using SHAP Values (Shapley Additive Explanations) calculated via the R package *SHAPforxgboost*[65]. Next, to generate environmental profiles for each snail, we constructed partial dependence plots (PDPs) using the R package *pdp*[66]. PDPs illustrate the average relationship between habitat suitability and the feature of interest. The average relationship is calculated by fitting models with all combinations of the non-focal variables for each value of the focal feature. To account for uncertainty in the model, feature contribution (SHAP Values) and PDPs were calculated 25 times, where, for each iteration, the model was trained on a randomly selected 80% of the data (bootstrapping). Data and code are available at https://github.com/ckglidden/biomphalaria-sdm-brazil.

### Hindcasting & counterfactuals

Our ultimate objective was to use historical information to estimate how *Biomphalaria* species distributions have changed with land-use and climate. To evaluate whether our model could track changes in

snail distribution through time, we trained the model on all data from 2000 to 2020 and tested the model on data collected from 1990 to 1999. We calculated model sensitivity, specificity, and AUC to evaluate how well the model could predict suitable snail habitats on historic out-of-sample data.

We then conducted a counterfactual analysis to examine how changes in climate and urbanization contributed to observed changes in *Biomphalaria* distribution over time. First, using the model trained on 2000–2020 data, we created prediction maps for the historical distribution of each snail (using data from 1992) and the current distribution of the snail (using data from 2017). We then created a climate counterfactual map by estimating model predictions using current data for all variables except climate, for which we used historical data (i.e., predicted what the current distribution would be if all variables but climate changed between the two time points). We repeated this process for urban variables to create an urban counterfactual. Finally, we compared the observed change in distribution between 1992 and 2017 with the observed change in distribution using the counterfactual scenario to estimate the change in distribution that was correlated with each set of features (i.e., climate or urban). For example, in a given location, if the change of probability of occurrence in the observed map is higher than change in probability in the counterfactual map (when climatology remained at historical values), then we infer that change in climatology is, at least in part, correlated with the increased probability observed between the two observed time periods. We created prediction maps for each of the 25 iterations described in the bootstrapping protocol above.

All figures were created using the R packages ggplot2[67], geobr[68], and raster[69].

### Reporting summary
Further information on research design is available in the Nature Portfolio Reporting Summary linked to this article.

## Data availability
Data is available on https://github.com/ckglidden/biomphalaria-sdm-brazil/ and via Zenodo using the doi 10.5281/zenodo.10975612. The repository contains the data used for training the SDM for each snail species ("glabrata_brt_data_april7.rds"; ("straminea_brt_data_a-pril7.rds"; "tenagophila_brt_data_april7.rds") as well as the data used to test model accuracy at predicting historical data ("glabrata_1999-test_data.rds"; "straminea_1999test_data.rds"; "tenagophila_1999-test_data.rds"). We obtained data from CHELSA, WorldPop (https://developers.google.com/earth-engine/datasets/catalog/WorldPop_GP_100m_pop), Global Human Settlement Layer (https://human-settlement.emergency.copernicus.eu/ghs_pop2019.php), MAPBIOMAS, JRC Global Surface Water Mapping Layers (https://developers.google.com/earth-engine/datasets/catalog/JRC_GSW1_4_GlobalSurfaceWater), Merit Hydro: Global Hydrography Dataset (https://developers.google.com/earth-engine/datasets/catalog/MERIT_Hydro_v1_0_1), NASADEM: NASA Digital Elevation (https://developers.google.com/earth-engine/datasets/catalog/NASA_NASADEM_HGT_001), WWF HydroSheds Free Flowing River Networks v1(https://developers.google.com/earth-engine/datasets/catalog/WWF_HydroSHEDS_v1_FreeFlowingRivers), OpenLandMap Soil Properties (clay: https://developers.google.com/earth-engine/datasets/catalog/OpenLandMap_SOL_SOL_CLAY-WFRACTION_USDA-3A1A1A_M_v02; sand: https://developers.google.com/earth-engine/datasets/catalog/OpenLandMap_SOL_SOL_SAND-WFRACTION_USDA-3A1A1A_M_v02; water: https://developers.google.com/earth-engine/datasets/catalog/OpenLandMap_SOL_SOL_WATERCONTENT-33KPA_USDA-4B1C_M_v01; carbon: https://developers.google.com/earth-engine/datasets/catalog/OpenLandMap_SOL_SOL_ORGANIC-CARBON_USDA-6A1C_M_v02; pH: https://developers.google.com/earth-engine/datasets/catalog/OpenLandMap_SOL_SOL_PH-H2O_USDA-4C1A2A_M_

v02; bulk density: https://developers.google.com/earth-engine/datasets/catalog/OpenLandMap_SOL_SOL_BULKDENS-FINEEARTH_USDA-4A1H_M_v02).

## Code availability
Glidden, CK, Singleton AL, Chamberlin A; Climate and urbanization drive changes in the habitat suitability of *Schistosoma mansoni* competent snails in Brazil; https://github.com/ckglidden/biomphalaria-sdm-brazil/; https://doi.org/10.5281/zenodo.10975595, 2024[70].

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

## Acknowledgements

This work was supported by the Belmont Collaborative Forum on Climate, Environment and Health (US-NSF ICER-2024383, FAPESP), by a grant of the Stanford Center for Innovation in Global Health, and the Stanford Program for Disease Ecology, Health and the Environment. G.A.D.L. was partially supported also by an NSF EEID grant (DEB – 2011179). E.A.M. and C.K.G. were supported by the National Science Foundation and the Fogarty International Center (grant no. DEB-2011147). EAM was additionally supported by the National Institute of Allergy and Infectious Diseases (grant nos R01AI168097 and R01AI102918), the National Institutes of Health (grant no. R35GM133439), and by seed grants from the Stanford Woods Institute for the Environment, King Center on Global Development, Center for Innovation in Global Health, and the Terman Award. C.K.G. was additionally supported by a Stanford Institute for Human-centered Artificial Intelligence Postdoctoral Fellowship. T.S.A. was supported by the National Institute of General Medical Sciences under grant number T32GM144273. S.H.S. was supported by NSF grant number 2024385. The content is solely the responsibility of the authors and does not necessarily represent the official views of the National Institute of General Medical Sciences or the National Institutes of Health. Our thanks to the Medical Malacology Collection from Fiocruz Minas (CMM-Fiocruz) for providing the *Biomphalaria* data.

## Author contributions

C.K.G., A.L.S., A.C., R.T., R.P., M.M., E.M., S.H.S., G.A.D.L. conceived and designed the analysis; R.T., R.P., R.L.C. collected the data; K.M.L., P.L., V.S., T.S.A. contributed data or analysis tools; C.K.G., A.L.S., A.C. performed the analysis; C.K.G., A.L.S., R.T., RP, MM, E.M., G.A.D.L. wrote the paper.

## Competing interests

The authors declare no competing interests.
