## [Peer Review File · Nature Communications]

Climate and urbanization drive changes in the habitat suitability of *Schistosoma mansoni* competent snails in BrazilREVIEWER COMMENTS

Reviewer #1 (Remarks to the Author):

This paper focuses on how climate change and urban development have influenced the habitat suitability of snails that are necessary for the transmission of Schistosomiasis, a tropical disease caused by Schistosoma parasites. The research utilizes machine learning, remote sensing, and extensive snail occurrence records to map the historical and current distribution of these snails across Brazil. Key findings include the impact of climate variables (such as temperature and precipitation) and urbanization patterns on snail distribution. The study's results are crucial for understanding changes in disease risk and developing effective control strategies for Schistosomiasis in Brazil. However, the authors could further supplement and clarify the following issues to enhance the quality of the article.

Introduction

1. Please expand on the historical context of Schistosomiasis in Brazil, highlighting previous control measures and their outcomes.
2. It's important to clearly articulate the specific research gap the study aims to address. The introduction should detail what aspects of Schistosomiasis modeling are yet to be explored or understood.
3. The objectives of the study might need to be more explicitly stated. A clear and concise statement of the research objectives helps to set the stage for the rest of the paper.
4. The introduction could further emphasize the relevance of the study to broader public health, environmental, and ecological issues. Linking the study to these larger themes can enhance its impact and relevance.

Methods

5. A more comprehensive description of how the model was validated, including the use of training and testing datasets.
6. Detailed explanation of the parameter settings for the chosen algorithm, which is crucial for replicability and understanding the model's performance.
7. In species distribution modeling, there are numerous algorithms such as Random Forest, Gradient Boosting Machine, and Support Vector Machine. However, this study exclusively utilized the XGBoost algorithm. One might question why multiple machine learning models were not employed to construct an ensemble ecological niche model to enhance the accuracy of the modeling.

Discussion

8. The discussion could benefit from a more detailed exploration of the broader implications of the findings. How do these results impact our understanding of Schistosomiasis transmission in other geographic areas or in relation to other diseases?
9. The discussion should offer a more comprehensive comparison with previous research. This includes how the findings align or differ from other studies and what these differences might mean for the field.
10. Suggest specific public health interventions or policy recommendations based on the study's findings, providing a clear link between the research and practical applications.
11. A more thorough acknowledgment and discussion of the study's limitations would strengthen the paper. This might include limitations in the data, the modeling approach, or external factors that could influence the results.
12. The paper could provide clearer guidance on future research directions. What are the next steps in this line of research, and what are the unanswered questions that future studies should aim to address?

Reviewer #2 (Remarks to the Author):

Please note I have made a few small comments on the main document and supplementary material which have been attached.

What are the noteworthy results?

The study found that there has been shifts in the distribution of schistosomiasis-transmitting snails across Brazil over the past three decades.

This was most prominently related to changes in land use, with peri-urban and urban environments as well as some small-scale agricultural activities that all provide more habitat to the snails that transmit schistosomiasis.

Will the work be of significance to the field and related fields? How does it compare to the established literature? If the work is not original, please provide relevant references.

This is important work as there is little published information on the current and potential future distribution of schistosomiasis in countries severely affected by it.

In comparison to established literature, this work provides original, new insights into schistosomiasis vector distribution in Brazil. It is also directly comparable to similar studies conducted in Zimbabwe, Uganda and South Africa in recent years that used similar methods to map the historic, current and potential future distribution of schistosomiasis transmitting snails in relation to climate change and anthropogenic impacts such as urbanization.

Does the work support the conclusions and claims, or is additional evidence needed?

The findings of the study directly support the conclusion and claims drawn.

However, it may be necessary to mention somewhere that the distribution of schistosomiasis vectors does not necessarily directly correlate to a known distribution of parasitic infections as the snails and parasites have different environmental needs. Understanding the distribution of the vectors is important to provide insights into where further research into the disease presence and potential mitigation strategies may be necessary. Still, it does not give a true reflection of the presence of the disease and infection rates in humans and animals.

Are there any flaws in the data analysis, interpretation and conclusions? - Do these prohibit publication or require revision?

The data analysis and interpretation thereof are sound and I do not see any flaws that prohibit publication thereof.

Is the methodology sound? Does the work meet the expected standards in your field?

The methodology is sound and meets the expected standards.

Is there enough detail provided in the methods for the work to be reproduced?

Enough detail is provided in the methods for the work to be reproducible.

Reviewer #3 (Remarks to the Author):

Glidden et al provide an important, distinctive perspective on changing distributions of *Schistosoma mansoni*-transmitting *Biomphalaria* snails in Brazil, based on machine learning, remote sensing, and informed by past snail surveys in Brazil. This has enabled them to generate new snail species distribution maps across large areas like Brazil and to assess what might happen in small 1km² areas too. Their novel results suggest that ranges of all three snail species shift, but that the shifts can mean no change in overall area occupied or increases or slight decreases in range area depending on the snail species. Insights regarding the degree of urbanization, particularly medium-sized urban areas, affecting distributions on local scales are also somewhat surprising and important and likely to have implications with respect to where schistosomiasis transmission might occur. I am impressed with the authors' ability to tease out the subtle shifts in range that seem to be occurring.

Overall, this approach seems powerful with potential applications to other species of schistosome snail hosts in other endemic areas, and I am enthusiastic to see it published. Of course, it remains to be seen how the approach will work when applied to forecasting rather than hindcasting, or how well it might work in areas without extensive long-term snail sampling to provide some sense of what is actually happening in the habitats themselves. And as the authors know, there are a host of biological factors that influence snail distributions on local or broad scales that may not show up in some remote sensing data.

I am not familiar with the suite of techniques used so I must rely on others to critically evaluate them.

Throughout, for the sake of public health people who may be most interested in, or in need of, the information provided, it would be helpful if a bit more emphasis could be placed in the main text clarifying some of the jargon and terms used. Some acronyms appear in the results before they are defined (definitions come later in the M&M section which may be less likely to be read) so providing the definition at first likely encounter for the reader would be helpful. Also, even brief mentions of what terms like "counterfactual analysis" mean in the context of their first appearance in the text would be helpful (at least for me).

For figures 1-3, some of the features indicated in part "a" like "upa" are not clear to me? With respect to "isothermality", and "temperature of the driest quarter", can something be said about the actual temperature values (air or water, time of day, etc) involved, to provide a bit more biological meaning for what these features are saying? That is, what temperature data are included in the remote sensing "climatological/environmental" data?

Is there anything the authors might briefly add about how their results might apply to the apparent invasiveness of *B. straminea* in places like China, or even of *B. tenagophila* in Europe? Disappearance or current rarity of *B. glabrata* from many Caribbean islands (biocontrol agents aside)?

One caveat to perhaps briefly mention is that *B. straminea* is part of a species complex (*straminea*-*kuhniana*-*intermedia*) and they are not easy to tell apart even for experts, and *kuhniana* and probably *intermedia* are not schisto vectors.

See manuscript for a few additional comments.

Response to reviewers: Climate and urbanization drive changes in the habitat suitability of *Schistosoma mansoni* competent snails in Brazil

Reviewer 1:

Reviewer #1 (Remarks to the Author):

This paper focuses on how climate change and urban development have influenced the habitat suitability of snails that are necessary for the transmission of Schistosomiasis, a tropical disease caused by *Schistosoma* parasites. The research utilizes machine learning, remote sensing, and extensive snail occurrence records to map the historical and current distribution of these snails across Brazil. Key findings include the impact of climate variables (such as temperature and precipitation) and urbanization patterns on snail distribution. The study's results are crucial for understanding changes in disease risk and developing effective control strategies for Schistosomiasis in Brazil. However, the authors could further supplement and clarify the following issues to enhance the quality of the article.

Thank you. We appreciate your positive comments and helpful suggestions.

Introduction

R1-1. Please expand on the historical context of Schistosomiasis in Brazil, highlighting previous control measures and their outcomes.

We added more context for schistosomiasis control in Brazil, including past control efforts and the country-specific recommendations for schistosomiasis prevention:

Line 71-79: "Schistosomiasis has historically been considered a rural disease. However, in the 1990s, at the start of our study, schistosomiasis began to emerge in some urban areas, such as the Recife Metropolitan area in Pernambuco, Brazil⁹. Since then, a growing portfolio of research in Brazil has demonstrated that *Biomphalaria* snails can thrive in human dominated environments within urban and peri-urban areas, such as drainage ditches, irrigation systems for small-scale agriculture, and unpaved flooded roads, all of which are often found in areas with marginalized populations and informal settlements with limited access to clean water, sanitation, and wastewater treatment¹⁰⁻¹⁵."

Lines 87-101: "Brazil aims for schistosomiasis elimination by 2030 and indeed has reduced transmission substantially since the inception of their national surveillance system in the 1950s¹⁶. However, total elimination remains elusive. Moreover, temporal trends in schistosomiasis related deaths have remained stable from 1999 to 2018 throughout most of the country, and has increased in the Northeast, where *B. glabrata* and *B. straminea* are found¹⁷. Globally, schistosomiasis control has been orchestrated through preventative and reactive medical treatment via administration of the drug praziquantel², WASH interventions (access to safe water, sanitation, and hygiene)², and environmental interventions (e.g., water engineering and snail control)^{18,19}. In Brazil, public health professionals have found that mass drug administration has temporary effects¹⁶. Therefore, current recommendations for risk reduction include integration of water sanitation, community-education, and removal of snail breeding sites through non-molluscicide based environmental interventions, such as aquatic vegetation removal, draining flooded areas, and modification of watercourses. Mass drug administration is only suggested under exceptional circumstances¹⁶."

R1-2. It's important to clearly articulate the specific research gap the study aims to address. The

introduction should detail what aspects of Schistosomiasis modeling are yet to be explored or understood.

We added the following text to the introduction:

Lines 110-111: “However, other environmental characteristics critical to snail habitat, such as land-use, have yet to be incorporated into snail habitat suitability maps.”

Line 121-124: “...but also to assess the functional form of the relationship between the environmental variable and habitat suitability (e.g, habitat suitability increases with precipitation seasonality)^{24,25}, which was not explored in previous studies mapping snail habitat suitability.”

R1-3. The objectives of the study might need to be more explicitly stated. A clear and concise statement of the research objectives helps to set the stage for the rest of the paper.

In lines 135-140 of the paper, we specifically outline the study objectives: “Specifically, by identifying environmental variables that most contributed to the probability of snail occurrence and using counterfactual analyses, we asked: (i) what are the key features of the environment, and their functional response, that influence snail habitat suitability? And (ii) how has snail habitat suitability shifted with climate and urbanization throughout the last three decades?”

To elaborate on our objectives, we added the lines:

Lines 140-143: “By doing so we develop a nation-wide consensus on the relationship between the environmental variables and snail habitat (e.g., determining if snail habitat is closer to rural or urban areas) and illuminate how dimensions of global change have and will shift targets for snail management.”

R1-4. The introduction could further emphasize the relevance of the study to broader public health, environmental, and ecological issues. Linking the study to these larger themes can enhance its impact and relevance.

Line 257-161: “In general, this study is one of the first to examine the compounding impacts of climate and land-use change on the distribution of infectious disease hosts at a national scale. Ultimately, it helps to predict the potential outcomes of interactions among local (e.g., urbanization) and large-scale (e.g., climate) environmental factors on environmentally transmitted disease risk.”

Methods

R1-5. A more comprehensive description of how the model was validated, including the use of training and testing datasets.

We included a more comprehensive description of validating the model in the main text:

Lines 538-546: “Spatial cross-validation (5-fold), using the R package *blockCV*⁶¹, was used to estimate the model's out-of-sample predictive power. Out-of-sample testing ultimately tests our model's ability to generalize biological patterns as opposed to the ability to predict complex patterns of the training dataset. We used spatial cross-validation so to minimize inflating model performance values due to spatial autocorrelation of environmental predictors⁶¹. In brief, we first split our dataset into five distinct geographic regions (5 folds). We then trained our model on four folds and validated the model (tested model performance) on the hold out fold. We repeated this process until all five geographic regions were tested as the hold-out dataset.”

R1-6. Detailed explanation of the parameter settings for the chosen algorithm, which is crucial for replicability and understanding the model's performance.

We updated the text to include parameters that were tuned during the model tuning procedure:

Lines 546-556: "For each spatial-cv step, model parameters were tuned and trained using Bayesian optimization via the R package *bayesianOptimization*⁶². Specifically, we tuned: the learning parameter ("eta"; controls how much information from a new tree will be used for boosting), maximum depth of the trees ("max_depth"), the minimum weight necessary to create a new node ("min_child_weight"), the fraction of data used to grow each tree ("subsample"), and the fraction of features used to train each tree ("colsample_bytree"). We tuned the model by using 5-fold cross-validation to select the hyperparameters that minimized the loss function. We controlled the balance of negative to positive weights by setting "scale_pos_weight" to 2 as there were 2x as many background samples (0) as presence samples (1). All other model parameters were set at default settings."

R1-7. In species distribution modeling, there are numerous algorithms such as Random Forest, Gradient Boosting Machine, and Support Vector Machine. However, this study exclusively utilized the XGBoost algorithm. One might question why multiple machine learning models were not employed to construct an ensemble ecological niche model to enhance the accuracy of the modeling.

A paper using a similar dataset in our lab compared the performance of MaxEnt, Random Forest, and Gradient Boosting Machines for modeling the distribution of *Biomphalaria* snails across different geographic extents. The manuscript found that predictions from GBMs best aligned with expert knowledge. We used Xgboost, a specific implementation of GBMs, since it is computationally faster than GBMs and better controls for over-fitting, especially with unbalanced datasets (ours has 2x the number of background points as occurrence points).

We included new text to clarify this point:

Lines 528 - 664: "Specifically, when compared to MaxEnt and Random Forest models, gradient boosted machine predictions of snail habitat suitability performed with comparable accuracy but better aligned with expert knowledge of *Biomphalaria*'s known distribution⁴¹."

Discussion

R1-8. The discussion could benefit from a more detailed exploration of the broader implications of the findings. How do these results impact our understanding of Schistosomiasis transmission in other geographic areas or in relation to other diseases?

Thank you for this comment. We included information on how these results impact our understanding of Schistosomiasis transmission in other geographic areas.

Lines 420-429: "Beyond Brazil, our study may help to inform spatial-temporal variation of snail habitat at a global scale. For instance, *B. straminea* in South China, where it is an invasive species, has demonstrated similar changes in distribution over time: the snail's distribution has expanded away from the equator towards historically more temperate habitats⁴⁰. Our results suggest that this pattern is likely due to changing climate, with localized suitability amplified or diluted by changing patterns in the extent of small to medium sized urban areas. Further, urban foci of transmission have emerged in Asia

and Africa, such as in Ethiopia and Nigeria¹⁰. Our study emphasizes the need to evaluate the role urbanization might be playing in schistosomiasis persistence on a global scale.”

R1-9. The discussion should offer a more comprehensive comparison with previous research. This includes how the findings align or differ from other studies and what these differences might mean for the field.

Thank you for this suggestion. We added important comparisons between our research and previously published literature.

Lines 318-331: “Our results for *B. tenagophila* mirrored the previous national model of the *Biomphalaria* climate envelope in Brazil: mean monthly temperature of the driest quarter was one of the top predictors of snail occurrence²⁰. However, our results contrast previously published results for *B. glabrata* and *B. straminea* as the previous national model found variables associated with temperature to be the strongest contributors to model predictions as opposed to precipitation²⁰. This may be related to the mismatch of *Biomphalaria* occurrence points and the climatologies used in previous studies, which were retrieved from WorldClim that spans 1960-2000³¹, whereas our climatologies overlap the decade the snail was collected. Yet, because this analysis is correlational, we cannot rule out the influence of unobserved variables that are correlated with these climatic conditions and may drive the discrepancy between the models. Further, it is difficult to comprehensively compare our results as the previous publication did not quantify the functional response between climate variables and probability of occurrence²⁰.”

Line 348-355: “However, as the location along the urban to rural gradient contributed more to model predictions than agricultural land-use and land-cover, there are likely other aspects of urban and peri-urban infrastructure that provide snail habitat, such as sewage systems that outflow into open ditches and unpaved streets in rapidly growing settlements³⁴, which also favor the dispersion in the environment of *S. mansoni* eggs with fecal waste^{27,34}. For example, ¹⁵ found that snails use sediment from used water and open-air sewage as a food source as opposed to aquatic plants.”

Lines 378-380: “...These findings are in line with local *Biomphalaria* surveillance, which have observed that *B. straminea* has replaced *B. tenagophila* in some areas of São Paulo state³⁵.

R1-10. Suggest specific public health interventions or policy recommendations based on the study's findings, providing a clear link between the research and practical applications.

We added the paragraph:

Line 402-418: “These findings may guide future management and policy work, such as allocating resources to control schistosomiasis in urban and peri-urban areas, and monitoring snail occurrence at the edge of the described climate envelopes. An important next step is identifying the mechanistic underpinnings of snail occurrence in peri-urban and urban settings, which will enable even higher precision mapping of snail habitat and the development of ecological interventions that facilitate sustainable snail removal. For example, in Senegal, ³⁸ found specific species of aquatic vegetation that

are strong predictors of *Bulinus* (the *Schistosoma* snail host in Senegal) abundance and schistosomiasis infection. Satellite imagery and deep learning models have been developed to identify this vegetation at < 2-meter resolution³⁹, providing a tool for highly targeted habitat control at a large geographic extent. Identifying fine-resolution biotic and abiotic drivers of snail habitat in peri-urban and urban Brazil will support similar precision mapping and resource-effective habitat removal. Importantly, *Schistosoma* parasites may have different environmental needs. Another necessary next step is to identify how snail habitat translates to human infection rates, and the environmental drivers of parasite presence that overlap with snail presence. This will enable identification of transmission hotspots as opposed to only snail habitat.”

R1-11. A more thorough acknowledgment and discussion of the study's limitations would strengthen the paper. This might include limitations in the data, the modeling approach, or external factors that could influence the results.

We appreciate this suggestion but feel as though we already address the key limitations of the study:

Lines 326-328: “Yet, because this analysis is correlational, we cannot rule out the influence of unobserved variables that are correlated with these climatic conditions and may drive the discrepancy between the models.”

Lines 355-360: “Critically, the population derived location along the urban to rural gradient is a rough proxy for estimating the impact of urban dynamics and should be updated in future studies to better incorporate specific infrastructure (e.g., waste management and resulting prevalence of organic pollution) and land cover composition of urban and peri-urban areas and estimate edge effects, especially as they may lag behind change in population density and size.”

Lines 391-398: “Notably, our analysis shows that there has been a decrease in suitability associated with the “disappearance” of small-medium urban areas between the time points. This warrants further on-the-ground investigation and validation as this result may be reflective of change in infrastructure associated with population size or might be an artifact of our definition of urbanization, where the population size and/or density only marginally drops below our urban population size and density thresholds, thus showing up on maps as a dissolved urban area when in reality there has been little change in infrastructure that impacts schistosomiasis risk.”

R1-12. The paper could provide clearer guidance on future research directions. What are the next steps in this line of research, and what are the unanswered questions that future studies should aim to address?

See comment R1-10.

Reviewer 2:

R2-1. It may be necessary to mention somewhere that the distribution of schistosomiasis vectors does not necessarily directly correlate to a known distribution of parasitic infections as the snails and parasites have different environmental needs. Understanding the distribution of the vectors is important to provide insights into where further research into the disease

presence and potential mitigation strategies may be necessary. Still, it does not give a true reflection of the presence of the disease and infection rates in humans and animals.

Thank you for this suggestion. We added this as an important future direction.

Line 415-418: “Another necessary next step is to identify how snail habitat translates to human infection rates, and the environmental drivers of parasite presence that overlap with snail presence. This will enable identification of transmission hotspots as opposed to only snail habitat.”

R2-2. Comments on supplementary materials.

We updated the figures legends in the supplementary materials to better describe the figures.

Reviewer 3:

Glidden et al provide an important, distinctive perspective on changing distributions of *Schistosoma mansoni*-transmitting *Biomphalaria* snails in Brazil, based on machine learning, remote sensing, and informed by past snail surveys in Brazil. This has enabled them to generate new snail species distribution maps across large areas like Brazil and to assess what might happen in small 1km² areas too. Their novel results suggest that ranges of all three snail species shift, but that the shifts can mean no change in overall area occupied or increases or slight decreases in range area depending on the snail species. Insights regarding the degree of urbanization, particularly medium-sized urban areas, affecting distributions on local scales are also somewhat surprising and important and likely to have implications with respect to where schistosomiasis transmission might occur. I am impressed with the authors' ability to tease out the subtle shifts in range that seem to be occurring.

Overall, this approach seems powerful with potential applications to other species of schistosome snail hosts in other endemic areas, and I am enthusiastic to see it published. Of course, it remains to be seen how the approach will work when applied to forecasting rather than hindcasting, or how well it might work in areas without extensive long-term snail sampling to provide some sense of what is actually happening in the habitats themselves. And as the authors know, there are a host of biological factors that influence snail distributions on local or broad scales that may not show up in some remote sensing data.

I am not familiar with the suite of techniques used so I must rely on others to critically evaluate them.

Thank you for your positive feedback and helpful suggestions.

Major comments:

R3-1. Throughout, for the sake of public health people who may be most interested in, or in need of, the information provided, it would be helpful if a bit more emphasis could be placed in the main text clarifying some of the jargon and terms used. Some acronyms appear in the results before they are defined (definitions come later in the M&M section which may be less

likely to be read) so providing the definition at first likely encounter for the reader would be helpful. Also, even brief mentions of what terms like “counterfactual analysis” mean in the context of their first appearance in the text would be helpful (at least for me).

Thank you for this suggestion. We defined model performance metrics (AUC, sensitivity, and specificity) and counterfactual analysis (see below). We checked the main text to make sure that methodological jargon was removed or defined. We defined the acronym SDMs earlier in the introduction and removed jargon associated with partial dependence plots in the results.

Lines 617-173: “Model performance was evaluated using sensitivity (i.e., the proportion of occurrences the model correctly identifies as occurrences), model specificity (i.e., the proportion of background points that the model correctly identifies as background), and model area-under-the-curve (AUC; i.e., a measure that calculates how well the model correctly distinguishes occurrence points from background points, where $AUC \leq 0.5$ indicates the model performs no better than a coin flip).”

Line 237-238: “...(CI: 52%-66%; CI is the 95% confidence interval estimated from a bootstrapping procedure)...”

Line 246-251: “We conducted a counterfactual analysis to quantify the change in habitat suitability associated with change in climate and urbanization. In brief, our counterfactual analysis compared the observed change in habitat suitability to the change in habitat suitability that would have occurred if the climate or urban extent had not changed. This methodology allowed us to isolate the change in habitat suitability associated with the change in these different dimensions of global change.”

R3-2. For figures 1-3, some of the features indicated in part “a” like “upa” are not clear to me?

Thank you for pointing this out. I updated figure axes to include the full name of each feature.

R3-3. With respect to “isothermality”, and “temperature of the driest quarter”, can something be said about the actual temperature values (air or water, time of day, etc) involved, to provide a bit more biological meaning for what these features are saying? That is, what temperature data are included in the remote sensing “climatological/environmental” data?

Thank you for this suggestion. I updated the text in a few different ways to increase biological interpretability (for isothermality and other focal climate variables):

- We updated the results to include longer descriptors of each important climate feature to better understand how each variable was calculated and their biological significance. For example, we changed “temperature of the driest quarter” to “mean daily air temperature during the driest quarter of the year”. Further examples in bold below.
- We updated the results to indicate the values and units associated with the relationship between the climate variable and change in snail habitat suitability. For example, I indicated that *B. tenagophila* habitat suitability decreased when mean daily air

temperatures during the driest quarter of the year exceeded 20°C, as opposed to only stating that *B. tenagophila* habitat suitability decreased with temperatures of the driest quarter. Further examples in bold below.

Lines 236-255: “For *B. glabrata* and *B. straminea*, we found that four out of the top five predictors of snail occurrence were climate variables, with three of these variables associated with precipitation (precipitation seasonality, precipitation in the driest month, and precipitation in the wettest quarter) and the fourth variable associated with temperature (isothermality) (Figure 1a, 2a). Our model indicates that, for both species, snail habitat suitability peaked at high precipitation seasonality (**12 CV, i.e., the coefficient of variance of monthly precipitation over one year**) but decreased in areas where, on average, **the mean monthly precipitation amount in the wettest quarter was high (< 6000 kg m⁻² year⁻¹)** (Figure 1b, 2b). For *B. glabrata*, habitat suitability also **decreased when precipitation amount was low in the driest month of the year (< 500 kg m⁻² year⁻¹)** (Figure 1b), while for *B. straminea*, habitat suitability **decreased when average annual precipitation amount was high (< 6000 kg m⁻² year⁻¹)** (Figure 2b). In regard to temperature, *B. glabrata* and *B. straminea* habitat suitability **non-linearly increased with isothermality, with respective peaks at 6 and 7°C (ratio of diurnal variation in relation to annual variation in temperature)**. In accordance with the known biology of *B. tenagophila*, which has lower thermal limits for survival and reproduction, the climate profile for *B. tenagophila* significantly differed from that of *B. glabrata* and *B. straminea*. Climate variables made up only two of the top five predictors (Figure 3a). *B. tenagophila* habitat suitability **non-linearly decreased with mean daily air temperature during the driest quarter of the year (>20°C) and peaked when precipitation seasonality is low (7 CV)** (Figure 3b).

As isothermality is hard to interpret, I also added a line to the discussion to provide an explanation for these results:

Line 311-314: “Further, *B. glabrata* and *B. straminea* habitat is most suitable at high values of isothermality -- with *B. straminea* habitat associated with higher isothermality than *B. glabrata* (Figure 1b, 2b). Isothermality is highest in northern Brazil, as such this variable might best reflect the unique climate conditions found towards equatorial Brazil.”

- Finally, I updated figure 1-3 to include units of measurement so better understand the biological relationships illustrated by the partial dependence plots.

R3-4. Is there anything the authors might briefly add about how their results might apply to the apparent invasiveness of *B. straminea* in places like China, or even of *B. tenagophila* in Europe? Disappearance or current rarity of *B. glabrata* from many Caribbean islands (biocontrol agents aside)?

Thank you for this suggestion. We have added the following paragraph:

Lines 420-429: “Beyond Brazil, our study may help to inform spatial-temporal variation of snail habitat at a global scale. For instance, *B. straminea* in South China, where it is an invasive species, has demonstrated similar changes in distribution over time: the snail’s distribution has expanded away from the equator towards historically more temperate habitats⁴⁰. Our results suggest that this pattern is likely due to changing climate, with localized suitability amplified or diluted by changing patterns in the extent of small to medium sized urban areas. Further, urban foci of transmission have emerged in Asia and Africa, such as in Ethiopia and Nigeria¹⁰. Our study emphasizes the need to evaluate the role urbanization might be playing in schistosomiasis persistence on a global scale.”

R3-5. One caveat to perhaps briefly mention is that *B. straminea* is part of a species complex (*straminea-kuhniana-intermedia*) and they are not easy to tell apart even for experts, and *kuhniana* and probably *intermedia* are not schisto vectors.

We appreciate that this may be a concern but *B. kuhniana* and *B. intermedia* are restricted to a few spots implying that they are much less adaptable to environmental conditions. We do think teasing apart ecological niches of each species could be an interesting future research project, but do not think the presence of the species complex impacts the interpretation of our results.

See manuscript for a few additional comments (answered below).

Introduction

R3-6. Could this approach also incorporate snail abundance?

Unfortunately, species distribution models are not typically used to quantify species abundance, particularly at this scale, as using species abundance necessitates systematic sampling, which is very hard to do at a national scale.

R3-7. Line 136- 137: Improve general interpretability of model validation & hindcasting performance (e.g., what is AUC)?

See comment R3-1.

R3-8. Line 154: Can temperature isothermality be associated with some idea of what values favorable temperatures actually seem to be?

See comment R3-3.

9. Line 160: accord with known biology of *B. tenagophila*?

We updated the sentence to identify that this aligns with the known biology of *B. tenagophila*.

Line 203-204: “In accordance with the known biology of *B. tenagophila*, which has lower thermal limits for survival and reproduction...”

10. Line 162: Some inkling of what specific temperatures are associated with decline?
See comment R3-2.

11. Line 209: What is a meso-region?

We included a definition of meso-region in the text (Line 264-265: “... i.e., an administrative level between municipality and state...”)

Discussion

12. its presence and persistence in China and Hong King areas?
See comment R3-4.

13. Line 262-265: Is there a possible sampling bias here towards areas close to towns?

Our background sampling methods account for sampling bias that arise from the concern that animals are generally sampled closer to human population centers, as these areas may be more amenable for sampling ecological communities. We added a supporting sentence to the methods to better illustrate this idea.

Line 474-474: “...such as bias that may arise if sampling of ecological communities is concentrated around human population centers.”

14. Line 269: is some index of organic pollution included among the variables tested?

It was not incorporated in this analysis as there are no indices of pollution available at the national level at fine spatial resolution, but this is important to consider in the future work. It has been added as a future direction.

Line 355-358: “Critically, the population derived location along the urban to rural gradient is a rough proxy for estimating the impact of urban dynamics and should be updated in future studies to better incorporate specific infrastructure (e.g., waste management and resulting prevalence of organic pollution)...”

REVIEWERS' COMMENTS

Reviewer #1 (Remarks to the Author):

The authors revised the manuscript according to our comments, and I have no further comments.

Reviewer #3 (Remarks to the Author):

I was favorably inclined towards the original manuscript and have looked through the author's responses to the reviewers' comments, and remain favorably inclined - the authors have done a thorough and convincing job in addressing the reviewers' concerns.